# Observation of the molecular response to light upon photoexcitation

Haiwang Yong [1], Nikola Zotev[2], Jennifer M. Ruddock[1,3], Brian Stankus[1], Mats Simmermacher[2], Andrés Moreno Carrascosa[2], Wenpeng Du[1], Nathan Goff[1], Yu Chang[1], Darren Bellshaw[2], Mengning Liang[3], Sergio Carbajo [3], Jason E. Koglin [3], Joseph S. Robinson [3], Sébastien Boutet[3], Michael P. Minitti [3], Adam Kirrander [2✉] & Peter M. Weber [1✉]

When a molecule interacts with light, its electrons can absorb energy from the electromagnetic field by rapidly rearranging their positions. This constitutes the first step of photochemical and photophysical processes that include primary events in human vision and photosynthesis. Here, we report the direct measurement of the initial redistribution of electron density when the molecule 1,3-cyclohexadiene (CHD) is optically excited. Our experiments exploit the intense, ultrashort hard x-ray pulses of the Linac Coherent Light Source (LCLS) to map the change in electron density using ultrafast x-ray scattering. The nature of the excited electronic state is identified with excellent spatial resolution and in good agreement with theoretical predictions. The excited state electron density distributions are thus amenable to direct experimental observation.

[1] Department of Chemistry, Brown University, Providence, RI 02912, USA. [2] EaStCHEM School of Chemistry and Centre for Science at Extreme Conditions, University of Edinburgh, David Brewster Road, Edinburgh EH9 3FJ, UK. [3] SLAC National Accelerator Laboratory, Menlo Park, CA 94025, USA.
✉email: adam.kirrander@ed.ac.uk; peter_weber@brown.edu

X-ray free-electron lasers (XFELs) are emerging as an important and powerful tool for research into the fundamental behavior of molecules and chemical dynamics[1–7]. The short duration, tunability, and extreme brightness of XFEL pulses allow for the application of sophisticated x-ray techniques in the ultrafast regime. Recent ultrafast non-resonant x-ray scattering experiments have demonstrated that it is possible to track structural changes during molecular vibrations[8] or chemical reactions, with analogous developments in ultrafast electron diffraction[9]. Non-resonant x-ray scattering probes the arrangement of electrons in the sample, and it has been suggested that it might eventually become possible to follow dynamic changes in the electron density upon photoexcitation experimentally[10–13].

Photoexcitation is the first step in all photochemical and photophysical processes, which include photovoltaics, photosynthesis, light-emitting diodes, photodynamic therapy, photocatalysis, and the primary events in human vision[14]. This first step results in a change in electron density that sets all subsequent dynamics in motion and ultimately determines the outcome of the reaction. Characterizing the initially excited electronic state is therefore an important task. The nature of the excited state has conventionally been inferred indirectly from spectroscopic measurements of transitions between states[15–18]. In terms of x-ray scattering, excited states have mainly been identified via secondary manifestations, such as the preferential alignment of a molecule with its transition dipole moment[19] or the changes in molecular geometry in an excited state intermediate[3,20–22]. Intriguingly, a recent x-ray scattering study demonstrated that in order to reproduce the correct coherent vibrational motion in an excited molecule, theoretical corrections that account for the change in electron density must be included in the data analysis[8]. The details of the changes in electronic structure, however, were obscured by comparatively large changes in molecular structure.

The experiment, shown schematically in Fig. 1, uses a low-pressure, room temperature gas of the molecule 1,3-cyclohexadiene (CHD). The CHD molecule serves as a model for many important reactions, including the synthesis of vitamin D in the skin upon exposure to sunlight[23], and the Woodward–Hoffman rules that contributed to the 1981 Nobel Prize in Chemistry. When excited to the 1B valence state by an optical laser at 267 nm, it undergoes a rapid electrocyclic ring-opening reaction, a process that has been captured by ultrafast x-ray[2] and electron scattering[24], as well as photoelectron spectroscopy[17,23,25] and x-ray spectroscopy[15]. In the present study, we use a higher-energy 200 nm pump pulse to excite the molecule to an electronic 3p Rydberg state. This carries a number of advantages in terms of the goals of our experiment. First, the 3p electronic state has a comparatively long lifetime of about 200 fs, as measured by photoelectron spectroscopy[26,27]. Second, the initial changes in molecular geometry are minor, which ensures that these do not obscure the redistribution of the electrons in the observed signal. This is further aided by the absence of electron-rich heavy elements in the molecule that could otherwise dominate the signal. Third, the diffuse nature of the excited 3p molecular orbital is markedly different from the highest occupied molecular orbital (HOMO), which provides additional confidence in the assignment of the electronic state in the scattering signal[12]. In the following study, we provide the direct evidence of the initial redistribution of electron density in real space upon photoexcitation.

## Results

**Experimental and theoretical results.** In the time-resolved x-ray scattering experiment, the ensemble of free CHD molecules is probed using 9.5 keV mean energy x-ray photons generated by the Linac Coherent Light Source (LCLS)[28], both with the excitation laser on and off. The excitation fraction is kept small to avoid effects from competing multiphoton excitation processes. The scattering signals are detected on a 2.3-megapixel Cornell-SLAC Pixel Array Detector (CSPAD) and binned according to the delay time between the laser pump and x-ray probe pulses. The detector images are decomposed into isotropic and anisotropic components[29] (see Supplementary Note 1). We focus on the isotropic rotationally averaged component that carries information on both the electronic and nuclear structure of the molecule. By using the fractional difference signal[30],

$$\Delta S(q) = \frac{I_{on}(q) - I_{off}(q)}{I_{off}(q)}, \tag{1}$$

where $q$ is the magnitude of the momentum transfer vector and $I_{on}(q)$ and $I_{off}(q)$ are the laser-on and laser-off signals, respectively, the effect of poorly defined experimental parameters such as background signals, gas pressure fluctuations, and pixel noise is minimized (see "Methods").

In Fig. 2a, we show direct evidence of the transfer from the tightly bound ground state to the diffuse 3p electronic state. The difference radial distribution function, $\Delta RDF(r)$, is obtained from the experimental difference signal at an early 25 fs pump–probe

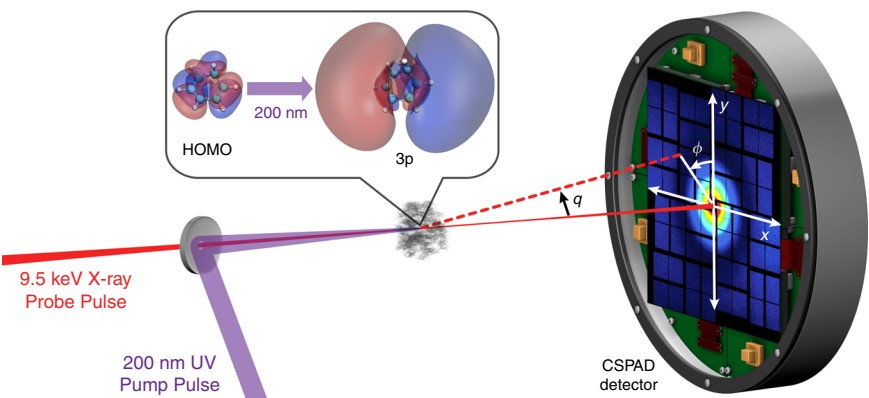

**Fig. 1 A schematic of the experimental set-up.** The CHD molecules are excited by a 200 nm UV pump pulse and the molecules are probed by 9.5 keV x-ray pulses with a variable time delay. The scattering signals are recorded on a CSPAD detector. The insert shows the highest occupied molecular orbital (HOMO), which has $\pi$ character, and the excited 3p molecular orbital. Both orbitals are rendered at 5% of maximum ISO values at the ground-state molecular geometry.

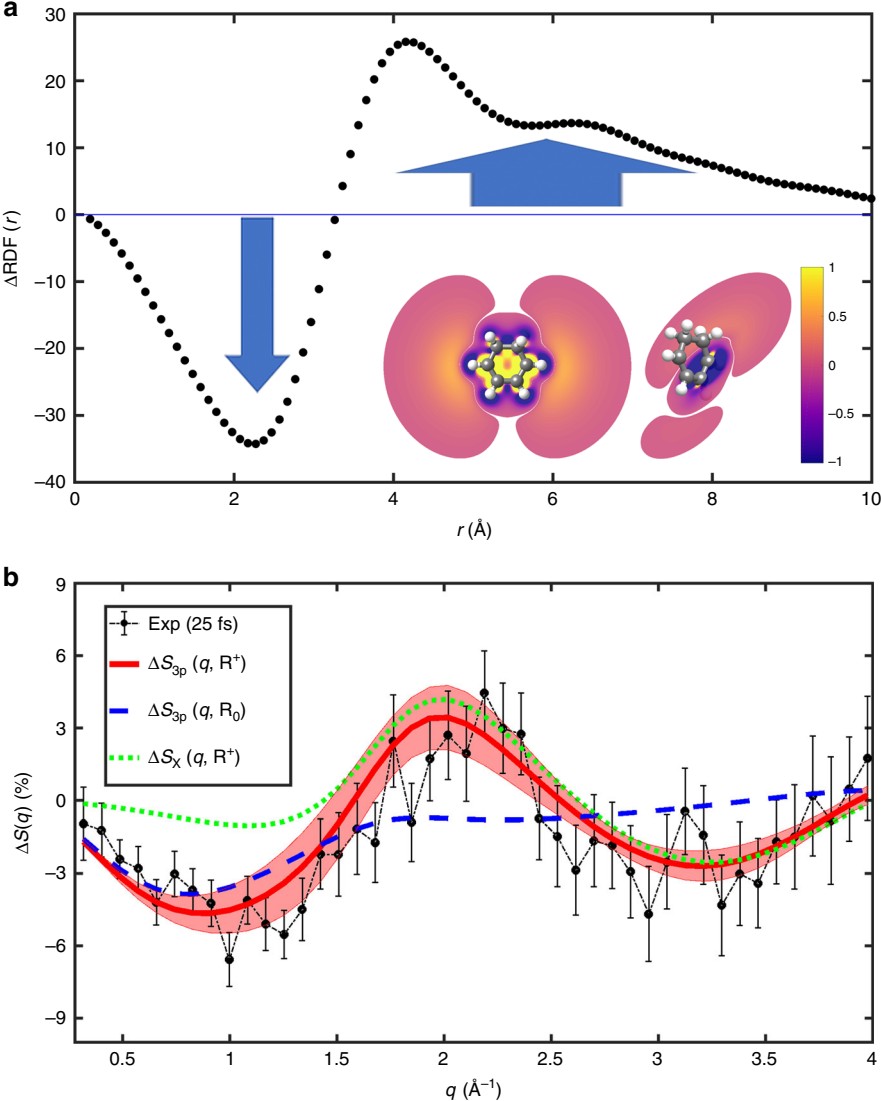

**Fig. 2 Experimental and theoretical signals. a** The real-space difference radial distribution function, $\Delta RDF(r)$, obtained from the experimental data at 25 fs pump–probe delay time. The blue arrows point to the depletion and increase in electron density at short and long electron distances, respectively, as the molecule is excited from the tightly bound ground electronic state to the diffuse excited 3p state. The insert shows the corresponding contour slices of the electron density difference from electronic structure calculations. The left-hand slice shows the difference in a plane through the C=C-C=C atoms, which illustrates the density gains far from the molecule, while the perpendicular right-hand slice, taken through one of the C=C bonds, shows the corresponding loss of density in the HOMO $\pi$-orbital. The color intensity is renormalized between −1 and 1 and absolute values <0.01 are not shown. **b** Fractional difference signals, $\Delta S(q)$, shown in percent. The experimental signal at 25 fs delay time is shown in black with $1\sigma$ error bars. The corresponding theoretical $\Delta S_{3p}(q, \mathbf{R}^+)$ signal for the electronic 3p state is shown in red with the shaded region accounting for the sampling of geometries in the excited state. For comparison, theoretical signals for the ground electronic state ($X$) at the 3p geometry, $\Delta S_X(q, \mathbf{R}^+)$, and for the excited 3p state at equilibrium geometry, $\Delta S_{3p}(q, \mathbf{R}_0)$, are included.

delay time via a sine transform (details in Supplementary Note 3),

$$\Delta RDF(r) = 2\pi^{-1} \int_0^\infty qr[I_{on}(q) - I_{off}(q)]\sin(qr)\mathrm{d}q. \quad (2)$$

It describes the difference in the probability distribution in real space of inter-electron distances before and after photoexcitation. The depletion of density at small distances, $r < 3$ Å, is matched by an increase in density in the interval 4–9 Å, verifying the diffuse character of the 3p excited state. In Fig. 2b, we show the experimental signal in the form of the fractional difference, $\Delta S(q)$, at 25 fs delay time. Experimental signals at other delay times (87 and 150 fs) are shown in Supplementary Fig. 3. The fact that experimental signals before 200 fs are almost identical indicates

that the molecular structure is stable upon optical excitation and that no coherent structural motion is observed at the current time resolution. By analyzing the time-dependent scattering signal integrated in two specific $q$ regions, 0.3–1.6 and 1.7–2.5 Å$^{-1}$, respectively (see Supplementary Note 2), we find a rapid onset in the small-$q$ region where the 3p electronic state features strongly, as discussed below, trailed by a slower onset of the scattering signal in the large-$q$ region where changes in the molecular structure are expected to appear. Finally, the scan of a wider range of delay times provides an excited state lifetime that agrees with the previous spectroscopic estimate of ~200 fs for the 3p state.

As an independent comparison, theoretical predictions for the ground and 3p electronic states are included in Fig. 2b. The theoretical fractional difference signal is derived under the

assumption that all dynamics occurs in the excited fraction (see Supplementary Note 5),

$$\Delta S_{exc}(q, \mathbf{R'}) = \frac{I_{exc}(q, \mathbf{R'}) - I_X(q, \mathbf{R_0})}{I_X(q, \mathbf{R_0})}, \qquad (3)$$

where $I_{exc}(q, \mathbf{R'})$ is the excited-state scattering at the perturbed molecular geometry $\mathbf{R'}$, and $I_X(q, \mathbf{R_0})$ the reference scattering from the ground-state $X$ at the equilibrium geometry $\mathbf{R_0}$. The scattering is calculated using ab initio multi-configurational wavefunctions obtained via second-order Complete Active Space Perturbation Theory (CASPT2)[31,32] (see "Methods"). The agreement between the predicted 3p signal $\Delta S_{3p}(q, \mathbf{R^+})$ and the experiment is excellent, even when taking into account the thermal structural distribution obtained by sampling geometries around the 3p state molecular geometry $\mathbf{R^+}$ (see Supplementary Note 4). The minor discrepancy between experiment and theory at large $q$ is mainly attributed to the fact that few photons are detected at the outer edges of the detector. Notably, the experimental signal in Fig. 2b is incommensurate with the corresponding signal for the electronic ground state, $\Delta S_X(q, \mathbf{R^+})$. Our assertion that small $q$ relates to changes in the electron density and large $q$ reflects changes in the molecular geometry is illustrated by the predicted signals also included in Fig. 2b. The signal for the 3p state at equilibrium geometry, $\Delta S_{3p}(q, \mathbf{R_0})$, is concentrated to small $q$, while the ground state at the 3p geometry, $\Delta S_X(q, \mathbf{R^+})$, mainly appears at large $q$.

**Decomposition of fractional difference signal.** This analysis can be extended by a careful examination of the fractional difference signal, $\Delta S_{exc}(q, \mathbf{R'})$, given in Eq. (3). By inserting a null contribution, $0 = I_X(q, \mathbf{R'}) - I_X(q, \mathbf{R'})$, this expression can be rewritten as the sum of two terms,

$$\Delta S_{exc}(q, \mathbf{R'}) = \frac{I_{exc}(q, \mathbf{R'}) - I_X(q, \mathbf{R'})}{I_X(q, \mathbf{R_0})} + \frac{I_X(q, \mathbf{R'}) - I_X(q, \mathbf{R_0})}{I_X(q, \mathbf{R_0})}$$
$$= \Delta S_{exc}^{elec}(q, \mathbf{R'}) + \Delta S^{nucl}(q, \mathbf{R'}), \qquad (4)$$

where the first term is the electronic contribution, $\Delta S_{exc}^{elec}(q, \mathbf{R'})$, defined via the difference in scattering from the excited and ground electronic states at a single molecular geometry $\mathbf{R'}$. The nuclear contribution, $\Delta S^{nucl}(q, \mathbf{R'})$, on the other hand, is indicative of the contribution to the scattering as if the molecular geometry was deformed $\mathbf{R_0} \rightarrow \mathbf{R'}$ on the ground electronic state. It should be emphasized that the decomposition of $\Delta S_{exc}(q, \mathbf{R'})$ in Eq. (4) does not involve any approximation regarding the scattering process itself.

In Fig. 3, we show the theoretical nuclear and electronic contributions to the percent fractional difference x-ray scattering signal, according to Eq. (4) above. In Fig. 3a, b, the magnitudes of the nuclear and electronic contributions are on the order of 4%, implying that changes in geometry and electronic state both contribute observably to the scattering signal. However, the electronic 3p state signal in Fig. 3b has a distinct negative signal in the low $q$ region (0–1.6 Å$^{-1}$), while the nuclear contribution is small in the same region. Contrary, the nuclear contribution grows for larger $q$ (1.7–2.5 Å$^{-1}$), a region where the electronic contribution only gives a small negative signal. This separation is related to the diffuse nature of the 3p Rydberg orbital compared to the HOMO. The scattering signal at very small values of the momentum transfer, $q \rightarrow 0$, is proportional to the square of the number of electrons in the molecule[33]. The signal for the molecular ion included in Fig. 3b drops toward −4.5% as $q \rightarrow 0$, corresponding to the removal of 1 of the 44 electrons in CHD. It remains largely parallel to the 3p state signal for $q > 1.0$ Å$^{-1}$,

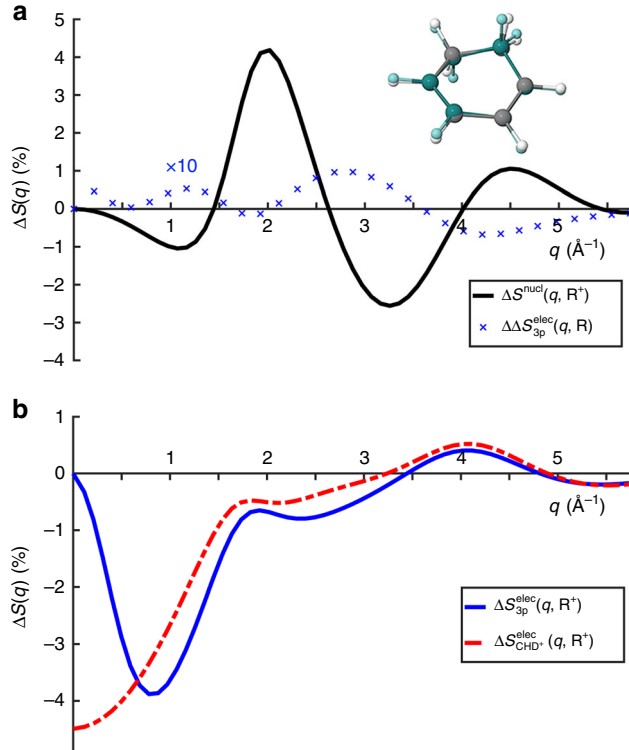

**Fig. 3 Separation of nuclear and electronic contributions.** Calculated fractional difference signals for the molecule CHD, assuming 100% excitation. **a** The nuclear contribution, $\Delta S^{nucl}(q, \mathbf{R^+})$, associated with the change in molecular geometry, $\mathbf{R_0} \rightarrow \mathbf{R^+}$, upon excitation (in black) and the small difference due to electronic effects, $\Delta\Delta S_{3p}^{elec}(q, \mathbf{R}) = \Delta S_{3p}^{elec}(q, \mathbf{R^+}) - \Delta S_{3p}^{elec}(q, \mathbf{R_0})$ shown at ×10 magnification (in blue). The insert shows an overlap of the molecule in the excited state 3p geometry ($\mathbf{R^+}$, gray) and the ground-state equilibrium geometry ($\mathbf{R_O}$, dark green). **b** The electronic contribution for the 3p state, $\Delta S_{3p}^{elec}(q, \mathbf{R^+})$, and for the molecular positive ion, $\Delta S_{CHD^+}^{elec}(q, \mathbf{R^+})$. Note that only the result for the 3p$_x$ state is shown since the 3p$_x$ and 3p$_y$ states have nearly identical signals.

implying that the 3p signal is dominated by the loss of the electron in the molecular core[8,12]. At $q > 3.5$ Å$^{-1}$, the electronic contributions of the ion and the 3p state are largely identical, suggesting that this region is mainly affected by the core electrons. Finally, Fig. 3a includes the difference in the 3p signal calculated at two geometries. The difference is only ~0.1%, at least one order of magnitude smaller than the other effects. This demonstrates that the electronic contribution in the 3p state is nearly independent of molecular geometry and suggests that the time evolution of the scattering signal in Rydberg states can be understood as arising from nuclear dynamics plus an approximately constant electronic contribution.

## Discussion

In summary, we demonstrate that ultrafast non-resonant x-ray scattering is capable of resolving changes in electron density due to transitions between electronic states. The experiment probes the rearrangement of electrons when gas-phase CHD molecules are optically excited from the ground electronic state to a low-lying Rydberg state. The current experiment is aided by the fact that CHD is a comparatively small organic molecule, consisting of light elements, and that the change in electronic structure is large while structural changes are small. However, given the upcoming improvements in XFEL repetition rate, time resolution, and mean photon energy[34] and the ongoing development of robust methods for data analysis[13,35,36], which may come to include sophisticated

tools from modern charge density analysis[37], it is clear that ultrafast x-ray scattering is set to become a powerful and versatile tool for chemical research. We anticipate that accurate measurements of the electron densities of ground and excited molecular states will provide key benchmarks for electronic structure theory, lead to a deeper understanding of how electron distributions evolve during the transformation of chemical bonds, and foresee experiments capable of monitoring the simultaneous structural and electronic changes in molecules during chemical reactions, thus providing unprecedented insight into chemical dynamics.

## Methods

**Time-resolved gas-phase x-ray scattering.** The experimental set-up has been described in detail previously[2,30]. The x-ray scattering measurements were performed at the CXI instrument[38] of the LCLS XFEL at the SLAC National Accelerator Laboratory. The optical pump laser was the fourth harmonic of a 120-Hz Ti: Sapphire laser operating at 800 nm, generating pulses at 200 nm with an ~80fs pulse duration and ~1 μJ/pulse on target. The x-ray probe pulses at 120 Hz repetition rate had pulse durations of ~30 fs and contained ~$10^{12}$ photons/pulse at 9.5 keV photon energy. The gaseous CHD sample pressure was controlled by a piezoelectric needle valve to ~6 torr of pressure at the interaction region. The gas cell and the detector are in vacuum, with an average background pressure outside the scattering cell of $2.6 \times 10^{-4}$ Torr, mostly comprised of CHD that flows out of the windowless scattering cell. The pulse energy and gas pressure were optimized for reduced background signal and <10% excitation probability. The pump and probe pulses were focused collinearly into the scattering cell, with approximate spot sizes of 30 μm full width at half maximum for the x-rays and 50 μm for the laser. The time delay between the pump and probe pulses was controlled by an electronic delay stage, and the timing jitter was monitored by a spectrally encoded cross correlator with a time resolution of 30 fs. In order to achieve the necessary noise level (<0.1%), the shot-to-shot x-ray intensity was monitored by a photodiode downstream of the scattering cell. The scattered x-rays were detected via a 2.3-megapixel CSPAD. Details of the detector calibration, the error analysis of the measured scattering signals as well as the decomposition into isotropic and anisotropic signals have been discussed in the Supporting Information of the article reported by Ruddock et al.[33]. The time-evolving scattering patterns extracted from the experiments can be expressed as a percent difference signal,

$$\%\Delta I(q,t) = \gamma \cdot 100 \cdot \frac{I_{on}(q,t) - I_{off}(q)}{I_{off}(q)} = \gamma \cdot 100 \cdot \Delta S \quad (5)$$

where $q$ is the magnitude of the momentum transfer vector, $\gamma$ is the excitation fraction, $I_{on}(q,t)$ is the isotropic scattering signal at delay time $t$ with the pump laser on, and $I_{off}(q)$ is the ground-state scattering signal with the pump laser off. The excitation fraction is a global parameter corresponding to the probability for the laser pulse to excite the molecules and is determined to be 6.0%. The fractional difference signals $\Delta S(q, \mathbf{R}')$ discussed in the main text are extracted from $\Delta I(q,t)$ by dividing out the excitation fraction and a time-dependent percentage value accounting for the time correlation between pump and probe pulse near time zero, calculated from the Heaviside step function convoluted by a Gaussian instrument function at chosen delay time points (see Supplementary Note 2 for more details).

**Ab initio calculations and scattering pattern simulations.** The geometry of the neutral CHD molecule in the electronic ground state was optimized using second-order CASPT2. The reference wavefunction was obtained from a Complete Active Space Self-Consistent Field calculation with four electrons in the four active valence $\pi/\pi^*$ orbitals performed with a double-zeta Dunning's correlation consistent basis sets augmented with diffuse functions, CASSCF(4,4)/aug-cc-pVDZ. In the same manner, the ionic ground state, used here to approximate the Rydberg state geometry, was optimized at the CASPT2 level after performing a CASSCF(3,4)/aug-cc-pVDZ calculation. For each of the two geometries, the ground and six lowest excited states are calculated in a multi-state CASPT2 calculation, whose reference wavefunctions originate from a state-average CAS(4,8)-SCF calculation with seven states. The orbitals included in the active space are again the four valence $\pi/\pi^*$ plus the 3s and 3p Rydberg orbitals. An improved description of the Rydberg character of the Rydberg orbitals is achieved by supplementing the aug-cc-pVDZ basis set by uncontracted 3s and 3p Rydberg basis functions positioned at the center of charge of the cation at the given geometry[39]. The character and energies of the first five low-lying states are given in Supplementary Table 1 and are in good agreement with prior experimental and theoretical results. All CASPT2 calculations employ a level shift of 0.3 Hartree to avoid intruder state problems. All ab initio calculations are performed using the MOLPRO electronic structure software package[40]. The elastic scattering patterns were then calculated using an in-house computational toolbox for analytical Fourier transformations of Gaussian-based electronic densities[31,32,41]. The inelastic contribution to scattering is approximated by tabulated atomic form factors[42]. To predict the theoretical percent difference scattering pattern in the 3p state as shown in Fig. 2, the percent difference signals for the $3p_x$

and $3p_y$ states are calculated separately using Eq. (3) and then combined with a ratio of 1:0.8. (The initial population of $3p_x$, $3p_y$, and $3p_z$ states for 200 nm excitation has previously been determined to be 1:0.8:0 from both an oscillator strength calculation[26] and an intensity analysis of photoelectron spectra[27]).

## Data availability
The data that support the findings of this study are available from the corresponding authors upon reasonable request.

## Code availability
The calculation of rotationally averaged elastic scattering patterns from ab initio wavefunctions has been discussed in earlier publications[31,32,35]. The code used for calculating scattering patterns, analysis of the raw experimental and simulation data, and for generation of the figures are available from the corresponding authors upon reasonable request.

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

## Acknowledgements

The authors thank G. Stewart (SLAC National Accelerator Laboratory) for his generous assistance with preparing the figures. This work was supported by the U.S. Department of Energy, Office of Science, Basic Energy Sciences under Award DESC0017995, the Carnegie Trust for the Universities of Scotland under research grant CRG050414 (to A.K.), a Carnegie PhD Scholarship (to N.Z.), the Royal Society of Edinburgh under Sabbatical Fellowship 58507 (to A.K.), and an EPSRC PhD Studentship from the University of Edinburgh (to D.B.). H.Y. acknowledges funding from the Brown University Global Mobility Research Fellowship and Brown University Doctoral Research Travel Grant to support research visits at the University of Edinburgh. Use of the Linac Coherent Light Source (LCLS), SLAC National Accelerator Laboratory is supported by the U.S. Department of Energy, Office of Science, Office of Basic Energy Sciences under Contract No. DE-AC02-76SF00515.

## Author contributions

P.M.W., A.K., and M.P.M. directed the project. M.L. and S.B. performed x-ray alignment and data collection. S.C., J.S.R., and M.P.M. performed laser alignment. J.E.K. provided software support during the experiment. W.D. and Y.C. performed record keeping during the experiment. H.Y., J.M.R., and B.S. performed analysis on the experimental data. N.Z., H.Y., A.M.C., and D.B. performed ab initio calculations and scattering pattern simulations. H.Y., N.Z., J.M.R., B.S., A.M.C., W.D., N.G., Y.C., D.B., M.S., A.K., and P.M.W. interpreted the results. H.Y., N.Z., M.S., A.K., and P.M.W. wrote the manuscript, in consultation with all other authors.

## Competing interests

The authors declare no competing interests.
