## [Peer Review File · Nature Communications]

Reviewers' comments:

Reviewer #1 (Remarks to the Author):

The authors have addressed my principle concerns. They have presented a reasonable case that ring opening does not make an appreciable contribution to the signal. I still think questions remain regarding the potential role of nuclear geometry in the measurement. While ring opening does not appear to matter, structural dynamics are involved in the transition to the electronic ground state and 6 eV of photon energy need to be converted to vibrational energy on the electron ground state potential. Nonetheless, I think it reasonable to consider these as questions for future work and I see no reason to impede publication at this point.

Reviewer #2 (Remarks to the Author):

The authors have very thoroughly addressed the concerns raised in the first review, sufficiently ruling out fundamental issues that might invalidate their claims. I would like to acknowledge and applaud the high quality and thoroughness of the responses. The authors have convincingly dealt with the concerns about the electronic component compensating for a misrepresentation of the geometric structure, as well as the discussion of the possible contributions from vibrations. In essence, I feel sufficiently convinced that their procedure works in principle, and would support publication in Nature Communication, if the below items can be addressed.

In general the scholarly presentation is of very high standard, but two issues were nevertheless disturbing the overall picture: (1) the authors state (p1 l. 19) that this is the 'first direct measurement of the ... redistribution of electron density ...', which is not obvious when some of the authors are also on their reference 10, which (as far as I can tell) also discusses the redistribution of electron density in a way that seems quite similar to the present study. (2) On p2. ll. 10-13 the authors go on to state that: " In terms of x-ray scattering, excited states have only been identified via ... the changes in molecular geometry in a long-lived excited state intermediate." This statement is not correct. Since at least 5-4 years a number of x-ray scattering studies have dealt with dynamic structures of excited states on the sub ps time scale, e.g. Biasin et al . PRL 2016, 013002, and their reference 4. I believe the findings of their ref 10 also contradicts this statement.

The potential impact is not convincingly demonstrated in the present manuscript. To truly demonstrate a transformative impact, it would be necessary to show that this is not just a coincidence found at 25 fs. In fact it seems quite surprising that the authors do not discuss the time evolution of the system, which they have clearly measured, as discussed in the SI. Judging from fig. 3 of their ref 10, they should be able to follow structural parameters as they (coherently) evolve after excitation. Since in this case the authors argue (p. 5 ll. 37-38) that the electronic configuration is "nearly independent of molecular geometry", plotting the structural and electronic contributions to the difference scattering signal at different times (and hence different nuclear coordinates in, what I would expect to be, a coherent motion of the excited molecule [re. their Ref. 10]), would provide strong support for their procedure (for separating nuclear and electronic components in the difference scattering signal) being valid.

Furthermore, the above would be necessary in order to to demonstrate capturing both the

structural and nuclear dynamic changes (as the impact is stated on (p. 1 l. 28-29)). What the paper presently demonstrates is a snapshot of the procedure applied at 25 fs, and not dynamics (which must have a time dimension). Furthermore, the demonstration seems to be limited to systems with light elements and very small geometric displacement, where the relative effect of the electronic signal can be expected to be at its most prominent. All in all, the paper shows that the procedure works as a proof of principle, but the general applicability is still to be shown. I realize that it might be impossible to include more time steps, but then it should at least be discussed why this is so and which implications this has for the conclusions that can be drawn from the study. Even in this case, the manuscript would still warrant consideration for Nature Communications, but the limitations outlined above should be considered in the manuscript.

We thank both Reviewers for their positive remarks and for recommending publication in *Nature Communications*. We provide full responses in the following, including minor changes in the manuscript where appropriate.

Reviewer #1 (Remarks to the Author)

The authors have addressed my principle concerns. They have presented a reasonable case that ring opening does not make an appreciable contribution to the signal. I still think questions remain regarding the potential role of nuclear geometry in the measurement. While ring opening does not appear to matter, structural dynamics are involved in the transition to the electronic ground state and 6 eV of photon energy need to be converted to vibrational energy on the electron ground state potential. Nonetheless, I think it reasonable to consider these as questions for future work and I see no reason to impede publication at this point.

We are grateful for this positive assessment and thank the reviewer for recommending publication of the manuscript.

With regard to the Reviewer's further remarks, the question of the potential role of nuclear geometry in the measurements has been addressed in the manuscript, and the SI clarifies the previous extensive responses to Reviewer 1. In terms of what happens with the molecular structure once the molecule has relaxed to its electronic ground state, this has no bearing on the current work but is indeed an interesting question that we have begun to address elsewhere (see e.g. Yong *et al. J. Chem. Phys.* **151**, 084301 (2019)). In the more general case, where the structural dynamics is more pronounced on the timescales reported, both structural reorganisation and electron density redistribution will have to be treated on an equal footing. This indeed something that we will work on in the future.

Referee #2 (Remarks to the Author)

The authors have very thoroughly addressed the concerns raised in the first review, sufficiently ruling out fundamental issues that might invalidate their claims. I would like to acknowledge and applaud the high quality and thoroughness of the responses. The authors have convincingly dealt with the concerns about the electronic component compensating for a misrepresentation of the geometric structure, as well as the discussion of the possible contributions from vibrations. In essence, I feel sufficiently convinced that their procedure works in principle, and would support publication in Nature Communication, if the below items can be addressed.

We greatly appreciate the Reviewer's careful reading of our manuscript and the positive assessment of our work. It is honored to see that our previous response is deemed of "high quality and thoroughness". We are grateful for the additional detailed comments and suggestions. We have revised the manuscript accordingly and provided a point-by-point response below where we address the remaining concerns. We hope that this should suffice to earn the Reviewer's support for publication in *Nature Communications*.

In general the scholarly presentation is of very high standard, but two issues were nevertheless disturbing the overall picture: (1) the authors state (p1 l. 19) that this is the 'first direct measurement of the ...

redistribution of electron density ...', which is not obvious when some of the authors are also on their reference 10, which (as far as I can tell) also discusses the redistribution of electron density in a way that seems quite similar to the present study.

Our Ref. 10 (now Ref. 8) discusses electron redistribution in an indirect manner, with the focus on its impact on our ability to measure coherent nuclear motion. In that paper, we observe long-lived vibrational motion in gas-phase N-methylmorpholine following 200 nm excitation using time-resolved x-ray scattering. When conducting the analysis of the data, we noticed that our observations could not be explained solely from the perspective of nuclear motion. Hence, we applied an *ad hoc* correction to account for the effect of electron redistribution in the manifold of the Rydberg excited states. In this sense, the work in Ref. 10 inspired the present study. The difference is, of course, that in the current work we measure the effect explicitly, by choosing a system that does not exhibit huge structural deformations. Going further, we are even able to show the extent of the electron redistribution in real space associated with the Rydberg state, as depicted in Figure 2a of the main text, and identify the excited electronic state. This constitutes a major step forward.

We have modified the Introduction with respect to previous work to ensure that the current work is understood in its correct context. We thank the Reviewer for the helpful and kind suggestion.

(2) On p2. ll. 10-13 the authors go on to state that: "In terms of x-ray scattering, excited states have only been identified via ... the changes in molecular geometry in a long-lived excited state intermediate." This statement is not correct. Since at least 5-4 years a number of x-ray scattering studies have dealt with dynamic structures of excited states on the sub ps time scale, e.g. Biasin et al . PRL 2016, 013002, and their reference 4. I believe the findings of their ref 10 also contradicts this statement.

The Reviewer is correct to point out that the 'long-lived' could be potentially misleading. We have removed the qualifier 'long-lived' and have added the relevant references. Thank you.

The potential impact is not convincingly demonstrated in the present manuscript. To truly demonstrate a transformative impact, it would be necessary to show that this is not just a coincidence found at 25 fs. In fact it seems quite surprising that the authors do not discuss the time evolution of the system, which they have clearly measured, as discussed in the SI. Judging from fig. 3 of their ref 10, they should be able to follow structural parameters as they (coherently) evolve after excitation. Since in this case the authors argue (p. 5 ll. 37-38) that the electronic configuration is "nearly independent of molecular geometry", plotting the structural and electronic contributions to the difference scattering signal at different times (and hence different nuclear coordinates in, what I would expect to be, a coherent motion of the excited molecule [re. their Ref. 10]), would provide strong support for their procedure (for separating nuclear and electronic components in the difference scattering signal) being valid. Furthermore, the above would be necessary in order to demonstrate capturing both the structural and nuclear dynamic changes (as the impact is stated on (p. 1 l. 28-29)). What the paper presently demonstrates is a snapshot of the procedure applied at 25 fs, and not dynamics (which must have a time dimension).

We agree with the Reviewer that being able to simultaneously observe both nuclear motion and changes in the electronic structure will be transformative. It is apparent that the Reviewer shares our passion for this ultimate goal. In recent years, ourselves and prominent other groups in this exciting area of research have demonstrated that it is possible to track the structural dynamics of a molecule. In this current work, we demonstrate that the second part of this vision is also achievable, *i.e.* that it is possible to observe and

identify an excited electronic state via the changes in the electron density upon excitation. Until now, it has been assumed that this will eventually be possible: our work constitutes a direct proof that this can be done. In order to further clarify that the current paper does *not* deal with nuclear dynamics, we have followed the Reviewer's suggestion and included experimental scattering data at two more time points along with their comparison to simulated signal (also shown in Figure 2b of the main text) in the SI as Supplementary Figure 3. From this Figure it is clear that we do not observe changes in structure during the first 200 fs, in agreement with our previous results from photoelectron spectroscopy. This gives us extra confidence that we indeed see the effect of the redistribution of electron density in excited state.

Clearly, the next steps in this line of research will be to convincingly combine both structural dynamics and electron dynamics. This is a complex problem and requires further advances both in experiments and data analysis. We are not there yet, but we insist that the current manuscript is an extremely important steppingstone.

We have added Supplementary Figure 3 in the SI and in the main text remarks that explain and interpret this additional data in the Results section of the main text.

Furthermore, the demonstration seems to be limited to systems with light elements and very small geometric displacement, where the relative effect of the electronic signal can be expected to be at its most prominent. All in all, the paper shows that the procedure works as a proof of principle, but the general applicability is still to be shown. I realize that it might be impossible to include more time steps, but then it should at least be discussed why this is so and which implications this has for the conclusions that can be drawn from the study. Even in this case, the manuscript would still warrant consideration for Nature Communications, but the limitations outlined above should be considered in the manuscript.

This is a nice point and the Reviewer is correct that careful elaboration of the limitations in the current study will only strengthen the results presented. We have therefore extended the Discussion in the manuscript to include considerations of how system size and other characteristics impact the observed signal. It is worth re-iterating that new facilities with higher repetition rate, such as LCLS-II and the European XFEL, are likely to have a transformative impact on both the signal-to-noise ratio and the resolution of the experiment. Thank you.

REVIEWERS' COMMENTS:

Reviewer #2 (Remarks to the Author):

In the revised manuscript, the authors efficiently and carefully address the points raised during the previous iterations. The work stands out as an important step in building a comprehensive analysis of structural and electronic effects in time resolved x-ray scattering for light element materials. This will only become more relevant with the new experimental capabilities offered by the European XFEL and the LCLS II upgrade. I fully support publication in Nature Commun.

Best regards,
Martin Meedom Nielsen